# Efficient Pan-Cancer Lesion Segmentation from Partially Labeled Data with nnU-Net

Yannick Kirchhoff[1,2,3], Maximilian Rokuss[1,3], Benjamin Hamm[1,4], Ashis Ravindran[1], Constantin Ulrich[1,4,5], Klaus Maier-Hein[1,6†], and Fabian Isensee[1,7†]

[1] German Cancer Research Center (DKFZ) Heidelberg, Division of Medical Image Computing, Heidelberg, Germany
[2] HIDSS4Health - Helmholtz Information and Data Science School for Health, Karlsruhe/Heidelberg, Germany
[3] Faculty of Mathematics and Computer Science, Heidelberg University, Heidelberg, Germany
[4] Medical Faculty Heidelberg, Heidelberg University, Heidelberg, Germany
[5] National Center for Tumor Diseases (NCT), NCT Heidelberg, A partnership between DKFZ and University Medical Center Heidelberg
[6] Pattern Analysis and Learning Group, Department of Radiation Oncology, Heidelberg University Hospital, Heidelberg, Germany
[7] Helmholtz Imaging, DKFZ, Heidelberg, Germany
`yannick.kirchhoff@dkfz-heidelberg.de`

**Abstract.** Accurate segmentation of cancer lesions in whole-body CT scans is essential for diagnosis and treatment planning. However, this task is challenging due to the diversity of lesion appearances and sizes, as well as the prevalence of partially labeled datasets. To address these challenges, Task 1 of the FLARE 2024 challenge was launched to encourage researchers to develop algorithms capable of generalized pan-cancer segmentation from a large, partially labeled dataset. In this paper, we describe our contribution to this challenge, utilizing nnU-Net with large batch size training and inference optimizations for efficient segmentation. Our best method achieved an average Dice Similarity Coefficient (DSC) of 15.6% and an average Normalized Surface Dice (NSD) of 17.3% on the validation set, with a mean inference time of 71.8 seconds and an area under the VRAM-time curve of 427,572 MB. Our second-best method achieved an average DSC of 13.5% and an average NSD of 13.8%, with a mean inference time of 44.9 seconds and an area under the VRAM-time curve of 224,872 MB. These results highlight the significant challenges inherent in pan-cancer lesion segmentation from partially labeled data under resource constraints, and underscore the need for further research in this area.

**Keywords:** FLARE challenge · Pan-Cancer Segmentation · nnU-Net.

---

† Shared last authorship

## 1   Introduction

Accurate organ and lesion segmentation in medical imaging is crucial to improve diagnostic accuracy, treatment planning, and monitoring the progression of diseases. Recent segmentation challenges in medical imaging have driven significant advancements in algorithm development, particularly for abdominal cancer segmentation; however, whole-body cancer segmentation presents unique challenges due to the wide variety of cancer types, lesion sizes, and anatomical regions in 3D CT scans.

Task 1 of the FLARE 2024 challenge builds on these developments, focusing on pan-cancer segmentation of primary as well as metastatic lesions. The provided dataset spans over 10,000 CT scans; approximately half of these have annotations for primary tumors only, while the rest have no annotations. The difficulty in this task mainly arises from handling missing labels in the partially labeled and unlabeled data splits, as well as the diversity of lesion appearance across different organs and cancer types.

Moreover, strict constraints on inference VRAM usage and time limit the possible network architectures, forcing careful trade-offs among model complexity, ensembling strategies, and test-time augmentations. This necessitates efficient models that can achieve high segmentation accuracy while remaining within resource limitations.

Learning from partially labeled data is an active area of research. A recent publication [7] approached this problem by introducing a *partial loss* formulation, which includes an *ignore class* for unlabeled parts of the image. This allows easy integration into state-of-the-art segmentation frameworks like nnU-Net [12]. However, this method requires partial labels for all classes—not only the foreground class—and is therefore not directly suitable in this setting.

Another promising strategy for handling missing labels is pseudo-label generation, which creates inferred labels for unlabeled data points based on the model's current predictions. This technique has been widely adopted in semi-supervised learning tasks, particularly for large unlabeled datasets. Notably, it was successfully implemented by the winners of the 2022 FLARE challenge [11,18], who employed pseudo-labeling to maximize performance on unlabeled datasets. The same method could be applied in this challenge to "fill in" missing labels in partially annotated scans.

This manuscript describes our approach for whole-body pan-cancer lesion segmentation from partially labeled training data in Task 1 of the FLARE 2024 challenge. We employ nnU-Net [12] with modifications for more efficient inference to adhere to VRAM and time constraints. Instead of specialized methods for handling partially labeled data during training, we rely on nnU-Net's foreground oversampling and a large batch size during training to effectively learn from the partial labels. We do not utilize the unlabeled subset for method development or training.

## 2   Method

Our contribution builds upon the state-of-the-art nnU-Net framework [12]. Due to the time and memory constraints imposed during inference, we cannot make use of the new ResEncL configuration [13] and even the smaller ResEncM configuration, similar in size to the default U-Net configuration, does not reliably adhere to the time limit. We therefore use the default nnU-Net configuration and only increase the batch size during training to account for the large dataset size and the difficulty of the task.

### 2.1   Proposed Method

**Preprocessing** All images are resampled to the median spacing of $[1mm, 0.8mm, 0.8mm]$ and normalized according to nnU-Net's CT-Normalization, i.e. intensity clipping to $[-900, 619]$ followed by subtracting $-52.2$ and dividing by $268.8$.

**Training:** nnU-Net generates a default configuration with a patch size of $96 \times 160 \times 160$, a batch size of 2, and a U-Net with 6 resolution stages. We use this default configuration but increase the batch size to 4 and 8 for two respective trainings. This larger batch size helps reduce false positive predictions and should make the training more stable against missing annotations in the partially labeled dataset. Figure 1 shows a schematic overview of the generated network architecture.

**Inference:** nnU-Net's inference pipeline is not optimized for single image inference as required in this challenge. We therefore make several small adjustments to the default pipeline to minimize resource usage and prediction time. First, we disable all test time augmentations, and calculate the argmax directly on the raw logits instead of the softmax probabilities. Second, we swap the default *skimage*-based resampling function for the much faster *torch* resampling, significantly speeding up segmentation export in exchange for a slight loss in performance.

## 3   Experiments

### 3.1   Dataset and evaluation measures

The segmentation targets cover various lesions. The training dataset was curated from more than 50 medical centers under license permissions, including datasets such as TCIA [3], LiTS [2], MSD [23], KiTS [8,10,9], autoPET [6,5], TotalSegmentator [24], AbdomenCT-1K [20], FLARE 2023 [19], DeepLesion [26], COVID-19-CT-Seg-Benchmark [17], COVID-19-20 [22], CHOS [14], LNDB [21], and LIDC [1]. The training set includes 4000 abdomen CT scans where 2200 CT scans have partial labels and 1800 CT have no labels. The validation and testing sets include 100 and 400 CT scans, respectively, which cover various abdominal

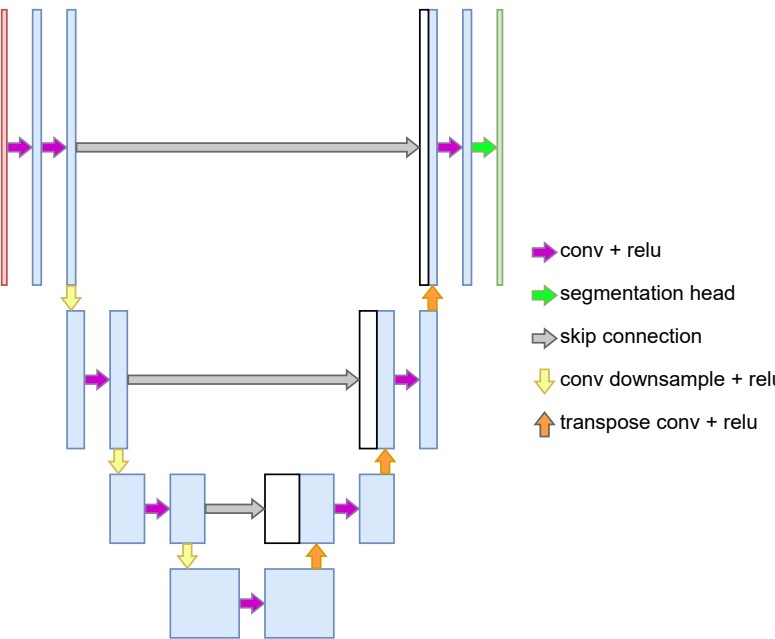

**Fig. 1.** Schematic network architecture of the U-Net created by nnU-Net's default configuration.

cancer types, such as liver cancer, kidney cancer, pancreas cancer, colon cancer, gastric cancer, and so on. The lesion annotation process used ITK-SNAP [27], nnU-Net [12], MedSAM [15], and Slicer Plugins [4,16].

The evaluation metrics encompass two accuracy measures—Dice Similarity Coefficient (DSC) and Normalized Surface Dice (NSD)—alongside two efficiency measures—running time and area under the VRAM-time curve. These metrics collectively contribute to the ranking computation. Furthermore, the running time and VRAM consumption are considered within tolerances of 45 seconds and 4 GB, respectively.

### 3.2   Implementation details

**Environment settings**  The development environments and requirements are presented in Table 1.

**Table 1.** Development environments and requirements.

| | |
|---|---|
| System | Ubuntu 20.04 |
| CPU | AMD Ryzen 9 3900X processor |
| RAM | 64GB DDR4-3600 RAM |
| GPU (number and type) | One NVIDIA RTX3090 GPU with 24GB VRAM |
| CUDA version | 12.1 |
| Programming language | Python 3.11 |
| Deep learning framework | torch 2.4.0 |

**Training protocols**  We train our models only on the partially labeled subset of the provided dataset, treating the partial labels as full labels. We rely on the large patch size and foreground oversampling to stabilize the training despite potentially conflicting signals. We use the default nnU-Net pipeline of data augmentations, consisting of spatial - i.e. rotations, mirroring - and intensity transformations, without further modifications. The final models were selected by expected inference times and performance on the public validation set. As all models predicted a significant amount of false positives on healthy patients, we neglect these for model selection and rely solely on best performance on pathological cases.

### 3.3   Test Set Submission

Task 1 of the FLARE challenge allowed for two submissions to the final test set. We therefore submitted a single model trained with a batch size of 8, which showed decent performance and efficiency on the public validation set (see Tables 3, 4). The second submission is an ensemble of two models trained with batch size 4 and 8, respectively. This provides better performance on the public validation set, but also increases inference time significantly (see Tables 3, 5).

**Table 2.** Training protocols.

| | |
|---|---|
| Network initialization | random |
| Batch size | 4/8 |
| Patch size | 96×160×160 |
| Total epochs | 1000 |
| Optimizer | SGD |
| Initial learning rate (lr) | 1e-2 |
| Lr decay schedule | PolyLR Scheduler |
| Loss function | Soft Dice loss + Cross Entropy loss |
| Number of model parameters | 101.94M |

## 4   Results and Discussion

### 4.1   Quantitative results on validation set

The results of the two methods submitted to the final test set on the public validation set are shown in Table 3. The ensemble of models trained with batch sizes of 4 and 8, respectively, improves by more than 2 dice points and 3.5 points in NSD upon the single model submission. Both submissions especially struggle on the LNDB cases in the public validation set with many false positive and false negative predictions, affecting the performance. This is also evident in the qualitative results presented in Section 4.2.

**Table 3.** Quantitative evaluation results of the two submitted methods on the public validation set. Results are further split into the two subsets, consisting of the FLARE part and the LNDB part.

| Methods | Public Validation | | Public Validation FLARE | | Public Validation LNDB | |
|---|---|---|---|---|---|---|
| | DSC (%) | NSD (%) | DSC (%) | NSD (%) | DSC (%) | NSD (%) |
| BS 8 | $13.5 \pm 24.7$ | $13.8 \pm 21.4$ | $39.7 \pm 35.2$ | $29.7 \pm 26.5$ | $7.8 \pm 17.1$ | $10.3 \pm 18.4$ |
| BS 4+8 | $\mathbf{15.6 \pm 25.7}$ | $\mathbf{17.3 \pm 22.9}$ | $\mathbf{40.8 \pm 35.1}$ | $\mathbf{30.5 \pm 26.3}$ | $\mathbf{10.1 \pm 19.1}$ | $\mathbf{14.5 \pm 21.0}$ |

### 4.2   Qualitative results on validation set

Figure 2 shows qualitative results of the submitted methods on four cases from the public validation set. The methods generally perform well on lesions in the abdominal region and sometimes on lung lesions. However, they are generally prone to errors on lung scans, predicting both false positives and false negatives in the majority of cases.

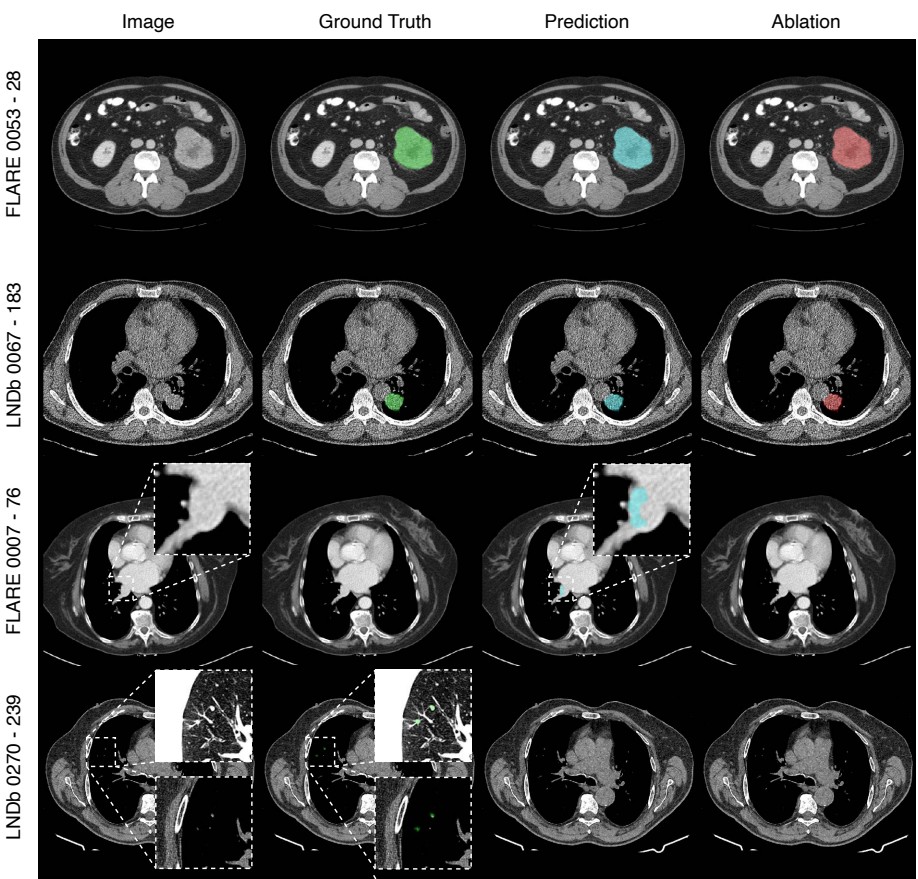

**Fig. 2.** Qualitative results of the two submitted methods on four example cases. *Prediction* denotes the prediction from the ensembled models, *Ablation* the single model prediction). The upper two rows show cases, where the models perform well, the lower down rows show examples of false positive and false negative predictions, respectively.

### 4.3   Segmentation efficiency results on validation set

Tables 4 and 5 show running time and VRAM utilization of both submissions
on 8 selected cases from the public validation set. The single model submission
exceeds the time limit only on one particularly large case, while the ensemble
submission exceeds the limit on 6 out of the 8 cases. Despite its better perfor-
mance (Table 3), the ensemble method will likely get a worse score on the final
test set due to the penalty for exceeding the time limit.

**Table 4.** Evaluation of segmentation efficiency of the single model in terms of the
running time and VRAM consumption of the single model submission. Total GPU de-
notes the area under VRAM-Time curve. Evaluation GPU platform: NVIDIA RTX3090
(24G).

| Case ID | Image Size | Running Time (s) | Max GPU (MB) | Total GPU (MB) |
|---|---|---|---|---|
| 0001 | (512, 512, 55) | 23.2 | 6678 | 120077 |
| 0017 | (512, 512, 150) | 42.0 | 6493 | 227860 |
| 0019 | (512, 512, 215) | 34.5 | 5762 | 150141 |
| 0029 | (512, 512, 554) | 73.8 | 6836 | 388930 |
| 0048 | (512, 512, 499) | 55.0 | 6945 | 257061 |
| 0051 | (512, 512, 100) | 42.8 | 6227 | 222682 |
| 0063 | (512, 512, 448) | 47.7 | 6738 | 233245 |
| 0099 | (512, 512, 334) | 40.3 | 6564 | 198982 |

**Table 5.** Evaluation of segmentations efficiency of the ensembled models in terms of
the running them and VRAM consumption of the ensemble model submission. Total
GPU denotes the area under VRAM-Time curve. Evaluation GPU platform: NVIDIA
RTX3090 (24G).

| Case ID | Image Size | Running Time (s) | Max GPU (MB) | Total GPU (MB) |
|---|---|---|---|---|
| 0001 | (512, 512, 55) | 39.0 | 5553 | 166657 |
| 0017 | (512, 512, 150) | 71.8 | 7136 | 404718 |
| 0019 | (512, 512, 215) | 48.8 | 6132 | 249758 |
| 0029 | (512, 512, 554) | 117.8 | 8894 | 813064 |
| 0048 | (512, 512, 499) | 81.3 | 8178 | 518122 |
| 0051 | (512, 512, 100) | 73.8 | 8177 | 507171 |
| 0063 | (512, 512, 448) | 75.7 | 8269 | 487778 |
| 0099 | (512, 512, 334) | 65.8 | 8219 | 423306 |

### 4.4   Results on final testing set

Tables 6 and 7 show the final results for segmentation performance and efficiency
on the test set, respectively. As expected from the results on the validation set,

**Table 6.** Segmentation performance on the test set.

| Methods | DSC (%) | | NSD (%) | |
|---|---|---|---|---|
| | Avgerage | Median | Avgerage | Median |
| BS 8 | $31.94 \pm 32.96$ | $18.93\,(0.22, 62.69)$ | $22.23 \pm 25.41$ | $13.07\,(0.00, 42.32)$ |
| BS 4+8 | $32.62 \pm 33.57$ | $18.00\,(0.03, 65.28)$ | $23.02 \pm 26.41$ | $12.22\,(0.00, 43.67)$ |

**Table 7.** Segmentation efficiency on the test set.

| Methods | Runtime (s) | | GPU (GB) | |
|---|---|---|---|---|
| | Avgerage | Median | Avgerage | Median |
| BS 8 | $44.71 \pm 17.20$ | $39.25\,(31.70, 57.56)$ | $92.77 \pm 40.24$ | $84.20\,(63.77, 114.20)$ |
| BS 4+8 | $60.43 \pm 19.93$ | $56.20\,(45.89, 75.26)$ | $157.07 \pm 69.90$ | $144.83\,(109.31, 183.75)$ |

performance for the ensemble model improves, but comes at the cost of longer runtime and higher VRAM consumption.

### 4.5  Limitation and future work

The main limitation of our work is the treatment of the partially labeled data. Large batch sizes and foreground oversampling alone do not seem to sufficiently stabilize the training but lead to subpar performance. Instead, it may make more sense to generate pseudo-labels based on initial training to retrain on the combined set of true labels and pseudo-labels. However, this approach introduces many false positive predictions, especially in regions like the lungs, as observed in the results sections. Therefore, additional steps would be required to ensure that the pseudo-labels actually improve performance rather than introducing noise. A better approach might be to build a genuinely partially labeled dataset that includes labels for definite background voxels. This could be achieved by exploiting additional information such as anatomical regions and involving pseudo-labels with appropriate post-processing.

Another approach to better utilize the partial labels without introducing conflicting signals for model training could be pretraining with a promptable model. We briefly experimented with such an approach, which showed very promising performance. However, transferring the pretrained weights to a non-promptable model version, and fine-tuning this model introduces further difficulties beyond the scope of this challenge. We therefore did not further pursue this idea.

## 5  Conclusion

In this paper, we addressed the challenge of resource-efficient pan-cancer lesion segmentation from partially labeled data in the context of Task 1 of the FLARE 2024 challenge. Our approach relied on training a default nnU-Net configuration with a large batch size to stabilize training. We submitted a method using only a single model with a batch size of 8 and a method employing an ensemble of two

models with batch sizes 8 and 4, respectively. The ensembling method increases performance notably but also leads to a significantly longer inference time, making it less competitive in this challenge. Despite the performance improvements from ensembling, both submissions still struggle with the correct segmentation of lesions, especially on lung CTs.

**Acknowledgements** The authors of this paper declare that the segmentation method they implemented for participation in the FLARE 2024 challenge has not used any pre-trained models or additional datasets other than those provided by the organizers. The proposed solution is fully automatic without any manual intervention. We thank all data owners for making the CT scans publicly available and CodaLab [25] for hosting the challenge platform.
The present contribution is supported by the Helmholtz Association under the joint research school "HIDSS4Health – Helmholtz Information and Data Science School for Health". Part of this work was funded by Helmholtz Imaging (HI), a platform of the Helmholtz Incubator on Information and Data Science. This work was partially supported by RACOON, funded by "NUM 2.0" (FKZ: 01KX2121) as part of the RACOON Project.

## Disclosure of Interests

The authors declare no competing interests.

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
