# OpenReview forum: "Efficient Pan-Cancer Lesion Segmentation from Partially Labeled Data with nnU-Net"
_MICCAI.org/2024/Challenge/FLARE — FLARE 2024 withMinorRevisions_

### Official Review · Reviewer_8M9w · 2025-01-26
**Review of ”Efficient Pan-Cancer Lesion Segmentation from Partially Labeled Data with nnU-Net“**

**Rating:** 8
**Confidence:** 4

**Review:**

In this paper, they propose an efficient segmentation method based on nnU-Net for pancarcinogenic transformation segmentation task in FLARE 2024 Challenge. By increasing the training batch size and optimizing the inference flow, the authors successfully achieve a better segmentation performance under resource constraints. The structure of the paper is clear, the experimental design is reasonable, and the results have certain reference value.
However, there is still room for improvement in the method when dealing with partially annotated data, for example, more efficient strategies for spury-label generation can be explored。

---

### Official Review · Reviewer_CrtM · 2025-01-27
**Efficient Pan-Cancer Lesion Segmentation from Partially Labeled Data with nnU-Net**

**Rating:** 8
**Confidence:** 4

**Review:**

This article introduces an efficient method for pan-cancer lesion segmentation using nnU-Net, which improves segmentation efficiency through large batch size training and inference optimization. The best method achieved an average DSC of 15.6% and NSD of 17.3% on the validation set, with an average inference time of 71.8 seconds and a VRAM-time curve area of 427,572 MB. Please add the slice number to Figure 2, e.g. Case #FLARETs_0001 (slice #123).

---

### Official Review · Reviewer_7aad · 2025-03-02
**minor issues in figures**

**Rating:** 8
**Confidence:** 5

**Review:**

1. Fig. 1 It would be better to specify the configs of the network (see the example in nnUNet supplementary).
2. torch resampling may lead to OOM issue for large CT scans (e.g., whole-body CT) on consumer GPUs (RAM<=16G). How did you address this issue?
3. Fig. 2 Please adjust CT images to proper window width and level
https://radiopaedia.org/articles/windowing-ct

---

> ### Author Response · Authors · 2025-03-29
>
> We adapted the points you raised. Regarding torch resampling, it is done on the CPU not on the GPU, but generally works faster than nnUNet's default resampling. We therefore don't have issues with OOM on the GPU and it actually doesn't use too much RAM on the CPU as well.

---

### Official Review · Reviewer_BFXj · 2025-03-12
**Review of "Efficient Pan-Cancer Lesion Segmentation from Partially Labeled Data with nnU-Net"**

**Rating:** 8
**Confidence:** 5

**Review:**

This research utilizes nnU-Net with large batch size training and inference optimizations for efficient segmentation.
1. Please adjust all CT images to the proper window width and level (e.g., 400/40 for abdomen CT).
2. Please explain the network layer represented by different color blocks in Fig 1.

---

### Author Response · Authors · 2025-03-29

We adapted the paper according to the reviewer comments, especially regarding window levels and width and clarity of figure 1.

---

### Decision · Program_Chairs · 2025-03-20

**Decision:**

Accept

**Comment:**

Please carefully address the reviewers' comments in the revision.